# Physical Activity and Its Diurnal Fluctuations Vary by Non-Motor Symptoms in Patients with Parkinson’s Disease: An Exploratory Study

**DOI:** 10.3390/healthcare10040749

**Published:** 2022-04-18

**Authors:** Koichi Nagaki, Shinsuke Fujioka, Hiroyuki Sasai, Yumiko Yamaguchi, Yoshio Tsuboi

**Affiliations:** 1Department of Neurology, Faculty of Medicine, Fukuoka University, Fukuoka 814-0180, Japan; nagaki@fukuoka-u.ac.jp (K.N.); shinsuke@cis.fukuoka-u.ac.jp (S.F.); yyamaguchi@fukuoka-u.ac.jp (Y.Y.); 2Research Team for Promoting Independence and Mental Health, Tokyo Metropolitan Institute of Gerontology, Tokyo 173-0015, Japan; sasai@tmig.or.jp

**Keywords:** Parkinson’s disease, physical activity, non-motor symptoms, pain and sensations, fatigue

## Abstract

Background: This exploratory study investigated the association between non-motor symptoms (NMS) and both physical activity and diurnal activity patterns in patients with Parkinson’s disease (PwPD). Methods: Participants included PwPD with modified Hoehn and Yahr stages 1–3. The presence of NMS was assessed with Movement Disorder Society-Unified Parkinson’s Disease Rating Scale (MDS-UPDRS) Part I. Physical activity was measured using a waist-mounted triaxial accelerometer. Logistic regression analyses evaluated associations between NMS and physical activity; furthermore, diurnal fluctuation in physical activity due to NMS was examined by ANCOVA. Results: Forty-five PwPD were included in the study. Among the domains of NMS, pain and other sensations (OR, 8.36; 95% CI, 1.59–43.94) and fatigue (OR, 14.26; 95% CI, 1.85–109.90) were associated with low daily step count (<4200 steps/day). Analysis by time of day showed no characteristic variability in physical activity but had constant effect sizes for pain and other sensations (*p* = 0.20, ES = 0.36) and fatigue (*p* = 0.08, ES = 0.38). Conclusion: Our exploratory study suggested that PwPD with pain and other sensations and fatigue recorded lower step counts than their asymptomatic counterparts. Therefore, PwPD with pain and fatigue may need more support in promoting physical activity.

## 1. Introduction

Parkinson’s disease (PD) is a progressive neurodegenerative condition that affects over 6 million people worldwide [1]. The incidence of PD is rapidly increasing in Japan, Europe, and other countries coincident with a growing aging population [2]. PD’s clinical characteristics include motor symptoms such as bradykinesia, tremor, rigidity, and gait disorders; and non-motor symptoms (NMS) such as hyposmia, constipation, cognitive impairment, pain, anxiety, depression, and sleep disorders [3,4]. Dopaminergic medications are effective in relieving some NMS (e.g., sleep disturbance and pain) as well as improving motor symptoms [5,6]. In contrast, some NMS such as psychosis, impulse control and related disorders (ICRDs), excessive daytime sleepiness (EDS), or constipation can also be worsened or even induced by dopaminergic medications [7]. Notably, non-pharmacological treatments can play an essential role in treating PD. In particular, exercise offers beneficial effects on both motor symptoms and NMS [8,9]. Physical activity has been shown to affect the brain’s neuroplasticity through the upregulation of brain-derived neurotrophic factor (BDNF) in rat models [10]. In turn, BDNF has been suggested to increase dopamine turnover in vitro and provide a neuroprotective role for dopaminergic neurons in the substantia nigra [11]. These underlying mechanisms may explain the clinical benefits of physical activity in alleviating both motor symptoms and NMS of PwPD.

It is well-known that the effectiveness of exercise depends on the stage of PD; the promotion of daily physical activity from an early stage of PD is most advantageous [12]. Integrating physical activity into the daily lives of PwPD is therefore one of the most effective and feasible approaches [13,14]. However, physical activity has been shown to decline over time in PwPD, even during the early disease stage when motor symptoms are less pronounced [15]. Insufficient physical activity is also associated with a decreased quality of life, an increased risk of falling, and a shortened life expectancy [16]. Although being involved in physical activity is an important means of disease management, many PwPD do not follow an exercise routine.

NMS appear before diagnosis and are common even in the early stages of the disease, including excessive daytime sleepiness, pain, fatigue, and mild cognitive impairment [17]. These symptoms affect the physical activity of PwPD from the early to advanced stages of the disease [18,19,20]. In order to minimize fluctuations in symptoms, exercise prescription and self-management are necessary, especially because diurnal variation in NMS can easily be overlooked by clinicians [21].

Previous studies in PwPD have often reported activity intensity as a measure of average daily physical activity [22]. Since many NMS fluctuate throughout the day, demonstrating their association with a daily average alone will not lead to their clinical application [23]. For example, early morning “OFF,” where motor symptoms worsen in the morning, can negatively affect quality of life (QoL) at a certain time of day [24]. No previous studies have tested whether there is a clear daily pattern of a decrease in physical activity among PwPD.

Several accelerometer-assessed activity metrics express physical activity, including step counts, time spent in light-, moderate-, or vigorous-intensity physical activity, and energy expenditure. When supporting physical activity in early PD, step counting plays a significant role as an instrument for behavioral change under self-management [25,26]. For this purpose, we believe that a commercially available wearable device or pedometer, which is inexpensive and easy to operate, allows self-monitoring, and provides a sense of accomplishment, and is a valid physical activity metric, especially among older PwPD. A previous study [27] reports that step count is most beneficial in assessing diurnal variation of physical activity and other factors in patients with PD and for exercise prescription.

Therefore, the primary purpose of this exploratory study was twofold: (1) to determine the association between the step count and the presence of NMS in PwPD and (2) to explore the difference in diurnal pattern (i.e., the time of day) of step counts among PwPD with and without NMS.

## 2. Materials and Methods

### 2.1. Participants

This cross-sectional study involved patients recruited at the Fukuoka University Hospital Parkinson Disease Care Center between February and March 2018. The PD diagnosis was based on the Movement Disorder Society (MDS) clinical diagnostic criteria [28]. Patients who had undergone complete physical and neurological examinations and a medical history review were included in this study. Exclusion criteria removed patients with cognitive decline as assessed by the Japanese version of the Montreal Cognitive Assessment; MoCA-J [29,30] of <21/30 points, modified Hoehn and Yahr (mHY) [31] stage IV or V, and those with an inability to walk independently. We also excluded patients with objective evidence of cognitive decline because we targeted patients who could self-manage their physical activity. There were no inclusion criteria based on the amount of physical activity. The study protocol was approved by the Fukuoka University Ethical Review Board (approval #2017m104). Patients or their legal representatives provided written informed consent.

### 2.2. Measurements

#### 2.2.1. Demographic and Clinical Characteristics

We collected data on age, gender, time since PD diagnosis, levodopa equivalent daily dose (LEDD), Movement Disorder Society-Unified Parkinson’s Disease Rating Scale (MDS-UPDRS) total score, mHY stage, and MoCA-J. The time since PD diagnosis was taken from the medical records. A total LEDD was calculated using established conversion factors for antiparkinsonian drugs [32]. The MDS-UPDRS Part III and the mHY stage were assessed through a physical assessment by movement disorder specialists. The MoCA-J is the scale of the first choice for measuring mild cognitive impairment and was administered by a clinical psychologist with specific training.

#### 2.2.2. Non-Motor Symptoms

NMS were evaluated using Part I of MDS-UPDRS [33]. MDS-UPDRS Part I has recently been validated against several other scales for assessing individual NMS [33]. Additionally, it has been compared with the non-motor symptoms scale, another validated instrument for assessing NMS as a whole in PwPD [34]. Therefore, we chose the MDS-UPDRS Part I because we believe it to be the most appropriate screening tool for individual NMS while capturing the overall picture of NMS. MDS-UPDRS Part I assesses the severity of the NMS experienced by the respondent over the past seven days. Each item (cognitive impairment, hallucinations and psychosis, depressed mood, anxious mood, apathy, feature of dopamine dysregulation syndrome, sleep problems, daytime sleepiness, pain and other sensations, urinary problems, constipation problems, and light-headedness on standing, and fatigue) was evaluated on a scale from 0 (normal) to 4 (severe). The total score (0–42) was used as an index of NMS severity (i.e., higher scores indicated severe NMS). Therefore, the MDS-UPDRS Part I score was based on the 13 sub-items classified by the presence or absence of symptoms (0: standard, 1–4) [35]. The MDS-UPDRS has good validity and reliability for screening PwPD and is translated into Japanese [36]. The cognitive function test of the MoCA-J and MDS-UPDRS Part I of the cognitive function item has different characteristics described as objective and subjective cognition. For example, even participants with a score of 21/30 or higher on the MoCA-J may subjectively experience cognitive decline, which is associated with future mild cognitive impairment [37]. Therefore, we considered it an assessment from a different perspective and used it in combination with the other tests.

#### 2.2.3. Physical Activity

Physical activity was evaluated using a validated hip-worn triaxial accelerometer (Active style Pro HJA-750C; Omron Healthcare, Kyoto, Japan) [38]. The Active style Pro HJA-750C is a useful measuring device as it shows an acceptable agreement with the internationally used ActiGraph GT3X+ (ActiGraph, LLC, Pensacola, FL, USA) with a 96.3% step count agreement rate [39]. The PwPD included in this study were deemed appropriate because, from their advanced stage, they showed a decrease in walking speed and stride length. However, Active style Pro HJA-750C did not underestimate the steps counted compared to the actual step count, even at a slower speed (55 m/min) [40]. The patients were instructed to continuously wear the accelerometer on the left side of their waist, except while sleeping. For safety reasons, they were also asked to remove the device when bathing or participating in contact sports. We used self-reported accelerometer wearing diaries and objective accelerometer measurements to determine when the accelerometer was not worn, such as during waking and sleeping hours. A recording was deemed valid if the accelerometer was worn for ≥10 h a day [41,42] for at least 3 days [43]. The records were expressed as step counts and metabolic equivalent (MET) for the minutes spent in daily life. Sedentary behavior (SB) was defined as 1.0 < MET ≤ 1.5, light physical activity (LPA) as 1.5 < MET < 3.0, and moderate-to-vigorous physical activity (MVPA) as 3.0 ≤ MET [44]. We used physical activity data from 7:00 h to 22:00 h only to standardize the diurnal pattern of physical activity.

### 2.3. Statistical Analysis

First, descriptive statistics were calculated by dividing the patients into two groups (< and ≥ 4200 steps/day) according to the clinical characteristics of PwPD. The reference value of 4200 steps was adopted because it is the first index based on the step count reported in PD and is easy for patients to use [45].

Second, a logistic regression analysis was conducted in the category with <4200 steps/day as the outcome variable and each subscore of Part I as the primary exposure variable. The logistic models were adjusted for age, gender, and mHY stage. Age is a necessary confounder in PD because the course of the disease may be different between patients with older and younger onset [46]. In terms of gender, physical activity has also been slightly higher in women [14]. In addition, mHY stages are reasonable confounders because physical activity is decreased from the early stage even under severe disease conditions. NMS also show an increasing trend with progression [17]. Because of the small sample size of this study, we prioritized confounders that were particularly important to make the analysis statistically conservative [47]. Since the independent variables were binary categorical data, the odds ratios became statistically unstable when the cell frequencies became small, so Cochran’s law was applied to *mutatis mutandis* or with necessary modifications. If the overall prevalence of each NMS item was <5, it was excluded from the analysis [48]. In addition, because this analysis included three confounders, we also excluded items with <3 one-sided cell frequencies.

For the sub-items for which significant differences were found in Part I, a power analysis was conducted as a post hoc test to visualize them as numerical values and to determine whether there were changes in the amount of physical activity over time. Finally, these items were subjected to a two-way repeated analysis of covariance (ANCOVA). We examined whether there was an interaction between the step counts taken with or without NMS and the time of day [morning (7:00–12:00 h), afternoon (12:00–17:00 h), and evening (17:00–22:00 h)]. The adjustment variables were the same as in the logistic regression analysis. Additionally, we showed the effect size of the interaction in a partial η2 [49]. All data were processed and analyzed using SPSS version 28 (IBM Corp., Armonk, NY, USA). A *p*-value < 0.05 was considered statistically significant.

## 3. Results

A total of 45 PwPD were enrolled in the study. The demographic characteristics of the patients studied are shown in Table 1. A total of 22 patients (48.8%) took fewer than 4200 steps, and 23 patients (51.1%) took more than 4200 steps/day.

Those who accumulated <4200 steps/day had significantly worse values for the mHY stage, LEDD, MDS-UPDRS total score, and MDS-UPDRS Part I score. Cognitive impairment, hallucinations and psychosis, depressed mood, anxious mood, apathy, features of dopamine dysregulation syndrome, and light-headedness on standing were excluded from the logistic regression analysis because their prevalence was less than five (Table 1). Logistic regression analysis of each of the MDS-UPDRS Part I scores showed that daytime sleepiness, pain, and other sensations (odds ratio (OR) = 8.36, 95% confidence interval of odds ratio (95% CI) = [1.59–43.94]), and fatigue (OR= 14.26, 95% CI= [1.85–109.90]) were associated in the crude model. In the subsequent adjusted model with age, gender, and PD severity (mHY) as confounders, only pain and other sensations and fatigue were associated with less than 4200 steps (no exercise habit) with no association for daytime sleepiness (Table 2). The fitting index by the Hosmer–Lemeshow test was pain and sensations (*p* = 0.54), and fatigue (*p* = 0.78). A comparison of step counts (continuous scale) by ANCOVA similarly showed that patients with pain and sensations, and fatigue items had significantly lower step counts (Appendix A). The associations between various activity indexes (i.e., MVPA, LPA, and SB) and NMS are also shown (Appendix A).

ANCOVA comparison of the NMS items associated with the step count showed significant differences in pain and other sensations and fatigue. In a two-way repeated ANCOVA with the time of day and presence of NMS as covariates for each subgroup, more patients in the group with NMS had step counts below 4200 steps/day for both pain and other sensations (*p* = 0.01) and fatigue (*p* < 0.01). It was not possible to show a characteristic trend in physical activity by time of day, but small effect sizes were shown for pain and other sensations (*p* = 0.20, ES = 0.36) and fatigue (*p* = 0.08, ES = 0.38). In a sub-analysis by ANCOVA of the presence of symptoms by time of day, the step count was significantly lower in the morning hours for participants with pain and other sensations (non-pain and other sensations, mean 2480.91 steps/time of morning, standard error (SE) = 260.34; with-pain and other sensations, mean 1514.57 steps/time of morning, SE = 266.54), but there was no significant difference between the afternoon and evening time periods. Those with fatigue had a significantly lower step count in the morning and afternoon hours (non-fatigue, mean 2286.76 steps/time of morning, SE = 251.93; with-fatigue, mean 1387.06 steps/time of morning, SE = 336.20), (non-fatigue, mean 1959.01 steps/time of afternoon, SE = 177.97; with-fatigue, mean 938.56 steps/time of afternoon, SE = 237.50; Figure 1), and there was no significant difference during the evening time period.

## 4. Discussion

This cross-sectional study examined the relationship between step count as a physical activity measured with an accelerometer and NMS as assessed by MDS-UPDRS Part I in mild to moderate PwPD. Pain and sensations, and fatigue were identified as NMS factors associated with PwPD having fewer than 4200 steps/day versus more than 4200 steps/day. These two factors suggested a characteristic difference in the step count at different times of the day. This study highlighted the potential value of designing physical activity interventions during the mild to moderate stages of PD without significant loss of motor function.

The association between physical activity and NMS shown in this study supports the hypothesis that physical inactivity is a factor in enhancing NMS in PwPD [50]. According to this hypothesis, reduced physical activity in daily life causes secondary exacerbations of PD motor symptoms and associated NMS, leading to a reduced capacity for physical activity. The amount of physical activity decreased with the severity of the disease, with a 21% decrease in HY stages 1 and 3 [14]. Consistent with the demographic information, there was a negative relationship between step count and disease severity: the more severe the condition, the lower the step count (Table 1).

In this study, pain and other sensations and fatigue were associated with non-exercise habits (fewer than 4200 steps/day); however, these symptoms are likely to appear in the premotor or early stages of PD [17,51]. Pain is poorly understood and remains essentially an expert opinion, even though it affects 85% of PwPD [52]. Studies evaluating the relationship between pain and other symptoms have not yielded consistent results. This is due to the diversity of pain symptoms in PwPD, including musculoskeletal nociceptive pain and central neuropathic pain [53]. PwPD with pain and other abnormal sensations generally had significantly lower physical activity than counterparts without such symptoms. In previous studies, the association between physical activity and pain has not been consistent [20]. Studies of patients with chronic pain have shown that increased physical activity can affect pain relief [54]. Exercise such as a planned, structured, repetitive, purposeful physical activity is recommended as part of a pain management program in PD, based on the evidence that exercise may reduce pain [55]. Nevertheless, the extent of beneficial pain reduction is dependent on the type and degree of pain and the intensity of the physical activity.

Analysis by time zone indicated that PwPD with pain and sensations tended to be involved in less physical activity, especially in the morning hours. Early morning “OFFs” are also highly prevalent in the early stages of PD. In a previous study, NMS was reported in 88.0% of early morning OFFs, with urinary urgency being the most common symptom, followed by pain and abnormal sensations in about half of the patients [56]. Therefore, it can be inferred that patients with pain symptoms in the morning hours will have reduced physical activity in the morning [24]. However, the time period with increased physical activity may differ depending on the type and degree of pain. A more detailed examination is needed to clarify how much exercise and at what time of the day is most suitable for specific types of pain.

Similarly to pain and other sensations, patients performing less than 4200 steps/day tended to have fatigue. A similar trend in the number of steps and moderate-to-vigorous activity indicated that patients with high physical activity did not feel fatigued, even if the causal relationship was unknown. Since many patients suffer from this symptom [57], an exercise routine may be one of the solutions to mitigate signs of fatigue.

Although the relationship did not reach a significant level, the observation of effect size suggests that there is a tendency for PwPD with fatigue to have diminished physical activity depending on the time period; a reason was that patients with advanced PD showed non-motor fluctuations (NMF) in addition to motor fluctuations (MF), and fatigue also fluctuated frequently [58]. Fatigue is more severe during an OFF phase, and diurnal fluctuation of physical activity may be associated with MF and NMF [58]. The OFF phase of MF appears after the disease has progressed to some extent, but fatigue may also appear early in the disease. In other words, the severity and variability of fatigue symptoms may occur as the disease progresses [59]. Recommended criteria for diagnosing fatigue in PD range from those that limit the type, intensity, or duration of the activity performed by the patient to those that show a diurnal pattern of intake regardless of the activity performed. Thus, fatigue may not exhibit a uniform pattern [60]. However, the power of the post hoc test to detect variation in fatigue and physical activity in this study was inadequate, and whether these types and patterns of fatigue change with severity or progression is not yet clear; a more quantitative collection of changes in fatigue may be needed to clarify the relationship between fatigue and physical activity in NMF. The best way to look at trends in fatigue at this time is to stratify in detail the severity of the disease, such as the HY phase.

This study has several limitations. First, using a hip-worn accelerometer excluded certain physical activities, including swimming, cycling, and upper body activities, and may therefore have underestimated total physical activity. Second, because the study was cross-sectional, we cannot draw causal conclusions about the effect of mild-to-moderate PD on physical activity. While our study suggests that variations in physical activity is not primarily driven by physical factors associated with mild-to-moderate PD, we did not directly test the hypothesis that NMS changes may be responsible: the more pain and fatigue a person has, the less they work; and as a result, the less active they are in the morning. Third, we suggest that physical activity values should be interpreted cautiously to reflect all PwPD because of the recruitment from a single institution, the limited number of participants in the study, and the undeniable possibility that some items were excluded due to the low prevalence of NMS. In addition, the possibility that the other NMS items are potentially related to the step counts cannot be ruled out. Fourth, the 4200 steps/day threshold is an estimate derived from the World Health Organization’s physical activity guidelines (adults with PD and other disabilities are recommended to participate in at least 150 min/week of moderate-intensity physical activity) [61], but should be interpreted with caution because there is not enough consensus. Future longitudinal and mechanistic studies based on these findings will help better elucidate how NMS and brain pathology associated with PD may directly impact physical activity patterns.

To more precisely characterize differences in daily physical activity between groups, we compared step count and diurnal pattern metrics derived from objective measures of physical activity that went beyond total average activity. These analytical methods provide a more detailed profile of how physical activity may decline during the early to advanced stages of the disease. Future studies exploring exercise habits and physical activity interventions among individuals with PD should consider measuring diurnal variations in physical activity.

The present study is the first to investigate physical activity in the early stages of PD using an objective measure of exercise habits and function. These results support studies of physical activity interventions that may help slow the disease course and preserve independent function among PwPD. It has been suggested that exercise intervention should consider the levels of physical activity and the activity time based on the health status of individual PwPD.

## 5. Conclusions

In the present study, PwPD performing fewer than 4200 steps/day and without good exercise habits tended to experience pain and sensations, and fatigue among NMS. This exploratory study was motivated by the possibility that activity levels may fluctuate according to the varying extent of pain and fatigue. However, the challenge of standardizing the multifarious nature of pain remained insurmountable in this first preliminary step. Categorizing pain based on type and intensity may help distinguish the specific symptom that could lead to decreased physical activity in PwPD.

## Figures and Tables

**Figure 1 healthcare-10-00749-f001:**
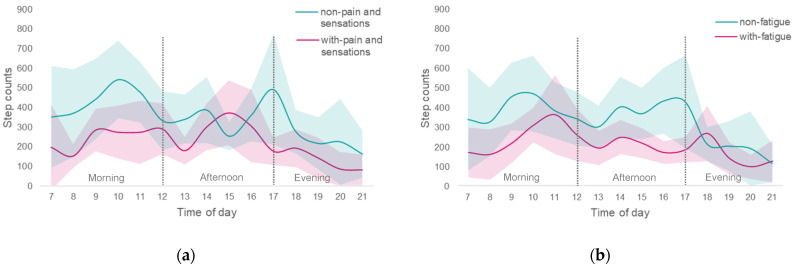
Comparison of step counts taken at different times of day with and without NMS; Red indicates patients with NMS, and green indicates patients without NMS. The solid line is the mean step count per time of day, and the surrounding light-colored zone is the 95% CI; (**a**) with and without pain and sensation; (**b**) with and without fatigue. There was no interaction by time of day for the pain and other sensations (*p* = 0.20, effect size by partial η2 (ES) = 0.36), and fatigue (*p* = 0.08, ES = 0.38) patient groups, respectively.

**Table 1 healthcare-10-00749-t001:** Demographic and clinical characteristics of the study patients.

Variables	Overall		Daily Step Count				
			<4200 Step/Day		≥4200 Step/Day		*p*-Value
	(*n* = 45)		(*n* = 22)		(*n* = 23)		
Age, years	63.8	(6.9)	63.6	(6.4)	64.0	(7.5)	0.85
Females, *n*	28	(62.2)	15	(68.2)	13	(56.5)	0.42
PD duration, months							
0–23	12	(26.7)	8	(34.8)	4	(18.2)	0.13
24–59	17	(37.8)	10	(43.5)	7	(31.8)	
>60	16	(35.6)	5	(21.7)	11	(50.0)	
mHY stage							
Stage 1	4	(8.9)	0	0	4	(17.4)	0.02
Stage 1.5	0	0.0	0	0	0	0	
Stage 2	16	(35.6)	5	(22.7)	11	(47.8)	
Stage 2.5	2	(4.4)	1	(4.5)	1	(4.3)	
Stage 3	23	(51.1)	16	(72.7)	7	(30.4)	
LEDD, mg/day	326.7	(132.1)	381.8	(135.0)	273.9	(107.5)	0.01
MoCA-J	25.6	(2.3)	25.6	(2.8)	25.6	(1.7)	0.98
MDS-UPDRS total score	40.9	(15.0)	46.6	(13.2)	35.4	(14.9)	0.02
MDS-UPDRS Part I score	4.0	(3.4)	5.5	(4.0)	2.5	(1.5)	0.01
MDS-UPDRS Part I sub-items (NMS) (%)							
Cognitive impairment	6	(13.3)	5	(22.7)	1	(4.3)	0.10
Hallucinations and psychosis	3	(6.7)	3	(13.6)	0	0.0	0.11
Depressed mood	7	(15.6)	5	(22.7)	2	(8.7)	0.19
Anxious mood	1	(2.2)	1	(4.5)	0	0.0	0.49
Apathy	4	(8.9)	3	(13.6)	1	(4.3)	0.35
Features of dopamine Dysregulation syndrome	1	(2.2)	1	(4.5)	0	0.0	0.49
Sleep problems	17	(37.8)	9	(40.9)	8	(34.8)	0.67
Daytime sleepiness	16	(35.6)	12	(54.5)	4	(17.4)	0.01
Pain and other sensations	22	(48.9)	15	(68.2)	7	(30.4)	0.01
Urinary problems	14	(31.1)	8	(36.4)	6	(26.1)	0.46
Constipation problems	27	(60.0)	14	(63.6)	13	(56.5)	0.63
Light-headedness on standing	2	(4.4)	1	(4.5)	1	(4.3)	0.97
Fatigue	17	(37.8)	13	(59.1)	4	(17.4)	<0.01
Physical activity							
Wearing time/day (min)	752.9	(95.5)	728.6	(99.4)	709.1	(98.9)	0.48
Step count/day	4552.7	(2496.2)	2433.9	(1084.2)	6579.3	(1599.0)	<0.01
SB, min/day	370.1	(81.4)	346.8	(83.8)	315.5	(75.9)	0.20
LPA, min/day	360.5	(116.7)	369.8	(109.5)	351.6	(125.0)	0.61
MVPA, min/day	27.3	(21.1)	12.0	(9.3)	42.0	(18.6)	<0.01

Values are shown as mean (SD) or *n* (%). PD, Parkinson’s disease; mHY, modified Hoehn and Yahr; LEDD, levodopa equivalent daily dose; MoCA-J, the Japanese version of Montreal Cognitive Assessment; MDS-UPDRS, Movement Disorder Society-Unified Parkinson’s Disease Rating Scale; SB, sedentary behavior; LPA, light physical activity; MVPA, moderate-to-vigorous physical activity.

**Table 2 healthcare-10-00749-t002:** Association between low step counts and non-motor symptoms by logistic regression analysis [dependent variable: low step count (<4200 steps/day)].

MDS-UPDRS Part I Sub-Items	Crude Model			Adjusted Model		
	OR	95% CI	*p*-Value	OR	95% CI	*p*-Value
Cognitive impairment	-	-	-	-	-	-
Hallucinations and psychosis	-	-	-	-	-	-
Depressed mood	-	-	-	-	-	-
Anxious mood	-	-	-	-	-	-
Apathy	-	-	-	-	-	-
Features of dopamine Dysregulation syndrome	-	-	-	-	-	-
Sleep problems	1.30	[0.39–4.34]	0.67	1.54	[0.36–6.49]	0.56
Daytime sleepiness	5.70	[1.45–22.35]	0.01	2.78	[0.60–12.77]	0.19
Pain and other sensations	4.90	[1.39–17.31]	0.01	8.36	[1.59–43.94]	0.01
Urinary problems	1.62	[0.45–5.78]	0.46	1.14	[0.25–5.12]	0.87
Constipation problems	1.35	[0.41–4.46]	0.63	0.97	[0.21–4.63]	0.97
Light-headedness on standing	-	-	-	-	-	-
Fatigue	6.86	[1.74–27.08]	<0.01	14.26	[1.85–109.90]	0.01

OR, odds ratio; 95% CI, 95% confidence interval of odds ratio; The adjusted model statistically controlled for age, gender, and mHY stage.

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
