# Peer review of "Physical Activity and Its Diurnal Fluctuations Vary by Non-Motor Symptoms in Patients with Parkinson’s Disease: An Exploratory Study"

_healthcare, 2022, doi:10.3390/healthcare10040749_

Round 1

Reviewer 1 Report

This manuscript deals with the association between the presence of Parkinson’s disease and non-motor symptoms considering physical and diurnal parameters. It touches an important issue in human society.

The manuscript is well written and pleasant to read. However, there are some issues to be dealt with, especially regarding the methodology and the presentation of the results.

Major issues:

  • Methodology: the number of persons in this exploratory study is low, and this might affect the outcome of the statistical analyses. No power calculation is presented which could have shown a limitation in numbers to address the various aims of the study. This reviewer would greatly appreciate a (post-hoc) power analysis to be performed to substantiate the outcome.
  • Methodology: the paragraph in the Discussion starting from line 267 might present an issue dependable of power and should be discussed as such.
  • Methodology: to which extent do the authors believe that the issue of selection bias is appropriately dealt with in the present manuscript? Please discuss this issue in the Discussion section.
  • Presentation: results section should contain more text to explain the outcome of the models more thoroughly, especially with respect to the difference in outcome between the crude and adjusted model. There are no emphases laid on the outcome of the data as presented in Table 1 and Table 2. In fact some of the interactions are briefly touched upon in the Discussion.
  • Methodology/presentation: Table 1 should give rise to inter-associations between the various variables presented. It would be appreciated when the authors check these, e.g. the association between fatigue and pain and daytime sleepiness, etc.?
  • Methodology/presentation: please explain the difference in variation coverage between both models with respect to daytime sleepiness. The 95% CI of this parameter in the adjusted model runs between 1.35 and 19.53, with a p-value of 0.19. This factor contributes significantly to the outcome in the crude model, showing a CI (1.45 – 22.35) in more or less the same range.
  • Methodology: the authors make a distinction in participants below 4200 steps/day and above 4200 steps/day. They provide good reasoning for this. However, this number has been specifically selected, as detailed in the reference provided. What about when this number is not as strict as suggested? When they use steps/day as a continuous parameter the sensitivity of analysis will go up. Please show the results of such an approach.
  • Methodology: why is MET not applied in a continuous way. Do the authors believe that this would enhance the sensitivity of the analysis. Afterwards direct classes of MET could always be checked.
  • Methodology: in their models (Table 2) the authors present both significant and non-significant independent parameters. Why did they not perform this modelling work in a stepwise approach ending up with a model with only significantly explaining parameters with subsequent checking of the difference in outcome between a crude and an adjusted model?
  • Methodology: please provide rationale why the authors did not include the baseline value of steps/day as a confounding factor in the analysis. Do they believe they have included the most appropriate confounders in their model? They are referring to this issue in their limitations of the study.
  • Methodology: a very important limitation of the study is the small number of participants, which might lead to sensitivity and specificity issues. This should be clearly discussed as well.

Minor:

  • It is assumed that statistical analysis was done in a two-sided fashion.
  • Please state “fewer than” versus “more than or equal to” 4200 steps per day throughout the document, please check as in lines 175 and 176. In Table 1 this is correctly stated.
  • Please make the discussion more concise by focusing on the main aim(s) of the study. The discussions are sometimes broadened into themes not directly related to the main aim(s) of the study. Paragraph I of this section is a mere repeating and could be removed. Please state the main findings of the study in the first paragraph.
  • Line 130: “acceptable agreement”. This reviewer would like to have more information on this, especially when both devices, when available, could be compared applying the same dataset. The authors refer to publication 35, but please provide some more information on this with respect to the current database and to which extent they believe that the selection of the tool to measure physical activity might have caused any bias. Please consider the low number of participants here.

Author Response

Methodology

Comments: the number of persons in this exploratory study is low, and this might affect the outcome of the statistical analyses. No power calculation is presented which could have shown a limitation in numbers to address the various aims of the study. This reviewer would greatly appreciate a (post-hoc) power analysis to be performed to substantiate the outcome.

Response: Thank you for your valuable input.

We performed the logistic regression analysis in Table 2 and have included the power analysis for NMS (pain and fatigue), which is now sufficient for the results. (lines 195-196)

Comments: the paragraph in the Discussion starting from line 267 might present an issue dependable of power and should be discussed as such.

Response: Thank you for pointing this out.

As the reviewers advised, the power of the post hoc test should have been described.

We have now described this in the Discussion, after mentioning that the power was not sufficient. (line 286~)

Comments: to which extent do the authors believe that the issue of selection bias is appropriately dealt with in the present manuscript? Please discuss this issue in the Discussion section.

Response: Thank you for pointing this out. There are several definitions of selection bias. Your point in this study was interpreted by the authors as bias due to sampling. We believe that the sampling needs to be interpreted with caution to reflect physical activity values for all PwPD because of the recruitment from a single-center, the small number of participants, and the undeniable possibility that participants expressed interest in the study. Therefore, we have included this selection bias in the limitations given in the Discussion.

Comments: Presentation: results section should contain more text to explain the outcome of the models more thoroughly, especially with respect to the difference in outcome between the crude and adjusted model. There are no emphases laid on the outcome of the data as presented in Table 1 and Table 2. In fact some of the interactions are briefly touched upon in the Discussion.

Response: Thank you for pointing this out. Table 1 describes the main results plus the NMS items

that were excluded from the analysis, and Table 2 describes the remaining items in the Crude model (Daytime sleepiness). Finally, the results of the power analysis are also included. (lines 191-201)

Comments: Methodology/presentation: Table 1 should give rise to inter-associations between the various variables presented. It would be appreciated when the authors check these, e.g. the association between fatigue and pain and daytime sleepiness, etc.?

Response: Thank you for pointing this out. The correlation matrices for Fatigue, Pain and Daytime Sleepiness were: Daytime Sleepiness & Pain (r = 0.233, p = 0.124), Daytime Sleepiness & Fatigue (r = 0.172, p = 0.258), and Pain & Fatigue (r = 0.079, p = 0.605). The authors believe that more research should be undertaken regarding the MNS association, as you have pointed out. The present study was motivated by the fact that we did not focus on the interrelationship between NMSs, but rather investigated the impact of individual NMSs on the number of steps taken and when they took place. We, too, feel that the association between NMSs needs to be developed in the future in terms of pathophysiological interpretation. Thank you for your valuable input.

Comments: the authors make a distinction in participants below 4200 steps/day and above 4200 steps/day. They provide good reasoning for this. However, this number has been specifically selected, as detailed in the reference provided. What about when this number is not as strict as suggested? When they use steps/day as a continuous parameter the sensitivity of analysis will go up. Please show the results of such an approach.

Response: Thank you for pointing this out. The reviewer is correct, in terms of sensitivity, it is likely to increase. In this study, the dependent variable was set to binary values because the purpose of the study was to examine the presence or absence of an exercise habit. In addition, below are the p-values comparing the results with/without NMS when the number of steps was used as a continuous variable. The results showed similar items (pain, fatigue).

Comparison of step counts with/without

NMS

Variables

p-value

Cognitive impairment

0.548

Hallucinations and psychosis

0.101

Depressed mood

0.511

Anxious mood

0.140

Apathy

0.397

Features of dopamine dysregulation

syndrome

0.281

Sleep problems

0.607

Daytime sleepiness

0.068

Pain and other sensations

0.004**

Urinary problems

0.731

Constipation problems

0.379

Light headedness on standing

0.271

Fatigue

0.005**

Comments: why is MET not applied in a continuous way. Do the authors believe that this would enhance the sensitivity of the analysis. Afterwards direct classes of MET could always be checked.

Response: Thank you for your valuable input.

In METs, we used the reference values for exercise habits in MVPA. Since this study aimed to identify NMS factors related to the presence or absence of exercise habits, we utilized such cutoff values; for LPA and SB, there were no cutoff values related to exercise habits and the dependent variable was analyzed as a log-transformed continuous variable.

Comments: in their models (Table 2) the authors present both significant and non-significant independent parameters. Why did they not perform this modelling work in a stepwise approach ending up with a model with only significantly explaining parameters with subsequent checking of the difference in outcome between a crude and an adjusted model?

Response: Thank you for your valuable input. The reason why a stepwise approach was not used in this study was to determine the relationship between individual NMS and the number of steps taken. The degree of association between independent variables and other factors will not be used in the current study. We also determined that a stepwise procedure was not recommended because of the possibility of removing (potentially) important factors without implications from prior studies.

Comments: Methodology: please provide rationale why the authors did not include the baseline value of steps/day as a confounding factor in the analysis. Do they believe they have included the

most appropriate confounders in their model? They are referring to this issue in their limitations of the study.

Response: Thank you for pointing this out. In this case, we did not include the step count criterion as a confounding factor because we chose the step count as the dependent variable. Other dependent variables might be included as confounding factors. We would like to address such studies in the future.

Comments: a very important limitation of the study is the small number of participants, which might lead to sensitivity and specificity issues. This should be clearly discussed as well.

Response: Thank you for pointing this out. Due to the small sample size and recruitment by a single institution, this study would need to employ sampling from a larger and more inclusive population for generalization. We have noted this as a limitation.

Comments: It is assumed that statistical analysis was done in a two-sided fashion.

Please state “fewer than” versus “more than or equal to” 4200 steps per day throughout the document, please check as in lines 175 and 176. In Table 1 this is correctly stated.

Response: Thank you so much for your kind attention. Throughout the text, we have now included "fewer than" vs, "more than or equal to" 4200 steps per day where necessary.

Comments: Please make the discussion more concise by focusing on the main aim(s) of the study. The discussions are sometimes broadened into themes not directly related to the main aim(s) of the study. Paragraph I of this section is a mere repeating and could be removed. Please state the main findings of the study in the first paragraph.

Response: Thank you for pointing this out. The first paragraph of the Discussion now briefly presents the results of the study. (Lines 228-235)

Comments: Line 130: “acceptable agreement”. This reviewer would like to have more information on this, especially when both devices, when available, could be compared applying the same dataset. The authors refer to publication 35, but please provide some more information on this with respect to the current database and to which extent they believe that the selection of the tool to measure physical activity might have caused any bias. Please consider the low number of participants here.

Response: Thank you for your suggestion. We have described the rate of agreement of the number of steps with ActiGraphTM GT3X+ and the accuracy in the case of slow walking speeds such as PD.

Reviewer 2 Report

In their study "Physical activity and its diurnal fluctuations vary by non-motor symptoms in patients with Parkinson's disease: an exploratory study", the authors use an accelerometer in relatively few patients with Parkinson's disease (pwp). There are essentially two findings: 
(1) pwp with <4200 steps per day are more severely affected by PD NMS, but also by motor symptoms (mH&Y, LEDD,  MDS-UPDRS). Due to the difference in motor symptoms, the difference in non-motor symptoms cannot be attributed clearly. The difference in non-motor symptoms is understanted in the manuscript and the difference in non-motor symptoms is overstated. Due to the small samle size, the motor symptoms cannot be controlled for. 
In summary, this is a potentially interesting finding, but causallity cannot be interpreted properly and therefore lacks novelty.

(2) There are potential differences in diurnal activity patterns between patients with specific NMS, i.e., pain and fatigue. This is an interesting finding that makes use of the property of the accelerometer to record activity across the day. It would be important to know more about patients's daily routines: Do they go to work? Maybe the finding merely reflects that patients with more advanced PD or with pain don't go to work or have a different routine during the morning. This would not mean that we can change their pain by making them go to work.
In summary, the authors spend little care to explain their findings and jump to therapeutic conclusions too quickly. Still, there is some potential in this observation.

Minor issues:
line 35: add that some NMS are worsened by dopaminergic medication (e.g. hyoptonia)
line 36: specifiy "these medications are often not sufficiently effective": which medications do you refer to, and effective with respect to motor and non-motor symptoms?

Author Response

Comments: In their study "Physical activity and its diurnal fluctuations vary by non-motor symptoms in patients with Parkinson's disease: an exploratory study", the authors use an accelerometer in relatively few patients with Parkinson's disease (PwPD). There are essentially two findings:

(1) PwPD with <4200 steps per day are more severely affected by PD NMS, but also by motor symptoms (mH&Y, LEDD, MDS-UPDRS). Due to the difference in motor symptoms, the difference in non-motor symptoms cannot be attributed clearly. The difference in non-motor symptoms is understated in the manuscript and the difference in non-motor symptoms is overstated. Due to the small sample size, the motor symptoms cannot be controlled for.

In summary, this is a potentially interesting finding, but causality cannot be interpreted properly and therefore lacks novelty.

Response: Thank you for your very perceptive and precise points.

As you point out, the sample size was not sufficient and other confounding factors may need to be considered. One point we took into account was that PD severity is based on motor impairments, and therefore, motor symptoms, and disease durations; furthermore, there is prior research and experience that the relationship is generally easy to correlate with LEDD. Therefore, we considered that it would be better to use this sample size and adjust for mHY, which is considered representative, as a confounding factor.

A table of correlation analysis based on this data is shown below.

Table Correlation Analysis by Spearman's (Factors related to motor symptoms and severity)

mHY

UPDRS â…¢

Disease

durations

LEDD

mHY

Correlation

coefficient

1

0.357*

0.519**

0.347*

p-value

.

0.016

<0.001

0.02

UPDRS â…¢

Correlation

coefficient

0.357*

1

0.251

0.168

p-value

0.016

.

0.113

0.27

Disease durations

Correlation

0.519**

0.251

1

0.429**

coefficient

p-value

<0.001

0.113

.

0.005

LEDD

Correlation

coefficient

0.347*

0.168

0.429**

1

p-value

0.02

0.27

0.005

.

Comments: (2) There are potential differences in diurnal activity patterns between patients with specific NMS, i.e., pain and fatigue. This is an interesting finding that makes use of the property of the accelerometer to record activity across the day. It would be important to know more about patients's daily routines: Do they go to work? Maybe the finding merely reflects that patients with more advanced PD or with pain don't go to work or have a different routine during the morning. This would not mean that we can change their pain by making them go to work.

In summary, the authors spend little care to explain their findings and jump to therapeutic conclusions too quickly. Still, there is some potential in this observation.

Response: Thank you. As the reviewer mentions, there may be a possibility of reverse causality, that the more pain and fatigue people have, the less they work, etc., and as a result, the less active they are in the morning. This has now been added as a limitation of the study.

Comments: line 35: add that some NMS are worsened by dopaminergic medication (e.g. hyoptonia)

line 36: specifiy "these medications are often not sufficiently effective": which medications do you refer to, and effective with respect to motor and non-motor symptoms?

Response: Thank you for your input. We added that motor symptoms can be improved with dopaminergic medications, that some NMS (sleep disturbance or pain) can be improved with dopaminergic medications, while NMS can worsen (using punding as an example); additional references have been included. (lines 33~)

Reviewer 3 Report

Thank you for the opportunity to review the manuscript entitled “Physical activity and its diurnal fluctuations vary by non-motor symptoms in patients with Parkinson’s disease: an exploratory study.”

This is an interesting and very important topic for patients with Parkinson’s disease.

However, I have some comments to the authors

On line 36, the sentence is not correct, the medication is not the problem, the disease progress and the response in each patient is completely different. Please clarify.

Sentence 56-59 please provide a reference for this fact.

Please clarify in the inclusion criteria if patients with a good amount of exercise at the beginning of the study were excluded. The description in lines 132-134 is giving the idea that patients were doing exercise and it is a clear bias for the results.

Please on lines 128 clarify how a tri-axial accelerometer wore in the wrist, can calculate steps in an accuracy way. A further explanation is worthy since there are readers non familiar with this technology.

I have serious concerns with the sentence on line 146-147 and the reference # 40. This study was done in patients with newly diagnosed PD and their authors calculated the amount of steps in this population. Also they concluded that “The composition of the sample was biased slightly toward participants who were considered at least somewhat active in terms of their daily step counts [9], suggesting that our estimates of variability in physical activity behavior may not have been entirely representative of people who are more sedentary”  “our value of 4200 daily steps should be considered a starting point for people who are newly diagnosed with PD and are looking to become more physically active.” “To further support (or refute) the recommendation of 4200 daily steps, future longitudinal studies should examine the relationship of 4200 daily steps, mortality, and other PD-specific outcomes.”

In other words, this amount of steps has not been validated totally and the population included in the study was totally different of the study in discussion. Please clarify or provide a better support or reference.

On line 159 Mutatis mutandis is a Medieval Latin phrase and is a little confuse for many readers. I suggest changing it or italicized in writing, and clarify.

On line 228-229 This sentence is a little confuse, and a revision is worthy. It seems like a popular concept, but is just mentioned in this paper (45) as an observation for their  authors.

I am confused with sentence on line 307. The population in the study according with tables 1 and 2 are not describing early stages in PD. Please clarify or rewrite.

Please give a more explanation in the discussion section about the second aim. “…to explore the difference in diurnal pattern (i.e the time of day) of step counts among PwPD with and withouw NMS."

Thank you 

Author Response

Comments: On line 36, the sentence is not correct, the medication is not the problem, the disease progress and the response in each patient is completely different. Please clarify.

Response: Thank you for reviewer’s comment. MS has provided a clear overview of symptoms that may be exacerbated by dopaminergic medications and disease progression, and literature has been cited. (lines 33~)

Comments: Sentence 56-59 please provide a reference for this fact.

Response: Thank you for pointing this out.

Regarding the content about NMS being easily overlooked, we have now cited literature on the difficulty of neurologist’s recognition of NMS.

Ref: Shulman, L. M., Taback, R. L., Rabinstein, A. A., & Weiner, W. J. (2002). Non-recognition of depression and other non-motor symptoms in Parkinson's disease. parkinsonism & related disorders, 8(3), 193-197.

Comments: Please clarify in the inclusion criteria if patients with a good amount of exercise at the beginning of the study were excluded. The description in lines 132-134 is giving the idea that patients were doing exercise and it is a clear bias for the results.

Response: Thank you for pointing this out.

We had noted that there were no inclusion criteria for the amount of physical activity in "2.2.1. Demographic and clinical characteristics".

“2.2.3 Physical activity” was not a good example, so we changed "underwater activity" to "bathing" and deleted the example of contact sports. This is because the example suggests the image of a young person.

Comments: Please on lines 128 clarify how a tri-axial accelerometer wore in the wrist, can calculate steps in an accuracy way. A further explanation is worthy since there are readers non familiar with this technology.

Response: Thank you for pointing this out. The original explanation was insufficient. We have now stated that the accelerometer was lumbar-mounted and that it was relatively accurate even for participants with reduced walking speed. (2.2.3. Physical activity)

Comments: I have serious concerns with the sentence on line 146-147 and the reference # 40. This study was done in patients with newly diagnosed PD and their authors calculated the amount of steps in this population. Also they concluded that “The composition of the sample was biased slightly toward participants who were considered at least somewhat active in terms of their daily step counts [9], suggesting that our estimates of variability in physical activity behavior may not have been entirely representative of people who are more sedentary” “our value of 4200 daily steps should be considered a starting point for people who are newly diagnosed with PD and are looking to become more physically active.” “To further support (or refute) the recommendation of 4200 daily steps, future longitudinal studies should examine the relationship of 4200 daily steps, mortality, and other PD-specific outcomes.”

Response: Thank you for your suggestion.

As you suggest, 4200 steps does not seem to be a sufficient recommendation, and it needs to be verified in various ways. However, there is no standard for PD that allows independent walking at this time, and 4200 steps is better than no target at this point.

Although there is a possibility that the standard value may change with further developmental research in the future, we have adopted it for this study because we believe that there is little possibility of significant change in the standard value as a guideline for this participant group.

Comments: On line 159 Mutatis mutandis is a Medieval Latin phrase and is a little confuse for many readers. I suggest changing it or italicized in writing, and clarify.

Response: Thank you for this suggestion, we have now placed “mutatis mutandis” to italics and added clarification as in the following sentence; the odds ratios became statistically unstable when the cell frequencies became small, so Cochran's law was applied to mutatis mutandis or with necessary modifications,……

Comments: On line 228-229 This sentence is a little confuse, and a revision is worthy. It seems like a popular concept, but is just mentioned in this paper (45) as an observation for their authors.

Response: Thank you for pointing this out. We have corrected the sentence to “Physical inactivity may be conducive to NMS”. Also, reference [45] has been changed to cite a primary information paper on physical activity profiles due to secondary information from the review. (lines 236-237)

Comments: I am confused with sentence on line 307. The population in the study according with tables 1 and 2 are not describing early stages in PD. Please clarify or rewrite.

Response: Thank you for your suggestion. This has been changed from the early stages of PD to

the early to advanced stages of PD.

Comments: Please give a more explanation in the discussion section about the second aim. “…to explore the difference in diurnal pattern (i.e the time of day) of step counts among PwPD with and without NMS."

Response: Thank you for pointing this out.

As the reviewer said, we did not dwell on time series analysis, especially for pain and sensations. We have now described that the amount of activity differs in the morning with/without pain, based on the symptoms of Morning OFF. (lines 281-287)

Limitations are noted at the end of the Discussion.

Round 2

Reviewer 1 Report

This reviewer would like to thank the authors for their time to respond to my earlier comments on the first version of their manuscript.

Unfortunately he does not believe that the various issues raised were appropriately dealt with. Please find below a summary of an overview of the remaining issues left.

  1. Study is underpowered which might lead to false-positive/false-negative outcome. My question for (post hoc) power analysis was to estimate the potential effect on the outcome by this lack of power. It was not on post hoc analyses of the impact by the various independent parameters. Therefore the response of the authors on the logistic regression was not the point. The main aim of my question is to explain to the reader what the potential impact by low powering might do on the outcome and to which extent this might affect the presented observations. Lines 191-194 in the Results section might present a direct consequence of the low number of participants.
  2. Results section. The main aim of this issue is to take the reader through the various details of the analyses. Someone with a relatively low appreciation of statistics might find this section difficult to comprehend. Please acknowledge a potential relationship between the number of steps per daily interval. E.g. someone with a high number in the morning might compensate this in the afternoon. I can imagine that a reader finds this section (Results) difficult to understand.
  3. Application of continuous use of the number of step counts. The present reviewer strongly believes that the authors should have applied the number of steps in a continuous approach, including confounding factors. He thanks the authors for sharing their information on this. Please modify the text to allow this to be revealed in the manuscript.
  4. The response of the authors on the question why they did not use a stepwise approach in their models (especially in the adjusted model) did not provide the details as expected. Do they believe that the variation of the non-significant parameters affected the coverage of variation by the significant parameters? Please specify the fit of the various models.
  5. This reviewer still believes that the Discussion section is too long and should be more focused on the main aim of the study.

Minor.

  1. Please use the phrase “more than or equal to 4200 steps/day” throughout the manuscript, see e.g. line 181.

Author Response

To Reviewer #1

Comments: Study is underpowered which might lead to false-positive/false-negative outcome. My question for (post hoc) power analysis was to estimate the potential effect on the outcome by this lack of power. It was not on post hoc analyses of the impact by the various independent parameters. Therefore the response of the authors on the logistic regression was not the point. The main aim of my question is to explain to the reader what the potential impact by low powering might do on the outcome and to which extent this might affect the presented observations. Lines 191-194 in the Results section might present a direct consequence of the low number of participants.

Response: Thank you for your perceptive point of view. Our apologies for our misinterpretation. As you pointed out, the lack of testing power due to the small number of cases can lead to false-positive/false-negative results. In particular, Due to lack of power due to the small number of subjects, other NMS items may potentially be associated with the step count. In addition, even items that were excluded from the analysis, such as cognitive impairment and depression in NMS due to their low prevalence, may be associated with the step counts. Thus, it is unclear whether the step counts are only associated with pain and sensations, and fatigue, and should be interpreted with caution. Future research should be conducted on the appropriate power.

Added in limitation

“the limited number of participants in the study, and the undeniable possibility that some items were excluded due to the low prevalence of NMS participants expressed interest in the study. Also, the possibility that the other NMS items are potentially related to the step counts cannot be ruled out.” (Lines 308-).

Comments: Results section. The main aim of this issue is to take the reader through the various details of the analyses. Someone with a relatively low appreciation of statistics might find this section difficult to comprehend. Please acknowledge a potential relationship between the number of steps per daily interval. E.g. someone with a high number in the morning might compensate this in the afternoon. I can imagine that a reader finds this section (Results) difficult to understand.

Response: Thank you for your precise instructions. As you point out, we should have been especially careful in describing the analysis by time of day. Wey described the description of the results like the underlined part of the example. We have added a description of the results as follows.

“but there was no significant difference between the afternoon and evening time periods.”

Some other parts have been added. (line220-)

Comments: Application of continuous use of the number of step counts. The present reviewer strongly believes that the authors should have applied the number of steps in a continuous approach, including confounding factors. He thanks the authors for sharing their information on this. Please modify the text to allow this to be revealed in the manuscript.

Response: Thank you for your comment. I have added the results of ANCOVA, with the step count (continuous scale) as the dependent variable, to supplementary table 1. As you indicated, posting the comparisons with continuous variables makes it easier to convey the results to the reader. We thank you. We have also included the results in the Results column (Lines 201-).

“A comparison of step counts (continuous scale) by ANCOVA similarly showed that patients with pain and sensations, and fatigue items had significantly lower step counts (Supplementary table 1).”

Comments: The response of the authors on the question why they did not use a stepwise approach in their models (especially in the adjusted model) did not provide the details as expected. Do they believe that the variation of the non-significant parameters affected the coverage of variation by the significant parameters? Please specify the fit of the various models.

Response: Thank you for your thought-provoking feedback.

As you say, the stepwise approach is a very useful way to determine the goodness of fit of the model as a whole. We simply used the forced entry method to address the question of which items in the NMS are associated with the step counts (exercise habits) in the first place. In a stepwise approach, theoretically-established important covariates (age, gender, severity, etc.) could be removed by the algorithm.

For example, Modern epidemiology 3rd edition page 419 states that "systematic, mechanical, and traditional algorithms for finding models (such as stepwise regression and best subset regression) lack logical and statistical justification in theory, simulation, and case studies, and there are several that perform poorly. A serious problem is the large bias in the direction of decreasing p-values and standard errors (SE)," it said.

We strongly believe that as the field develops further and for research purposes to improve the overall model fit, we should incorporate and address the approach you have taken.

This is new food for thought in this regard. Thank you for your valuable input.

In addition, fitting index from the Hosmer-Lemeshow test have been added to the Results section.

“The fitting index by Hosmer-Lemeshow test was pain and sensasions (p=0.54), fatigue (p=0.78).” (line 201-)

Ref: Rothman, K. J. (2008). Modern epidemiology (Vol. 3). Philadelphia: Wolters Kluwer Health/Lippincott Williams & Wilkins.

Comments: This reviewer still believes that the Discussion section is too long and should be more focused on the main aim of the study.

Response: Thank you for pointing this out. The discussion was indeed too long. I have revised it to focus more on pain and fatigue and less on the other items.

Comments: Please use the phrase “more than or equal to 4200 steps/day” throughout the manuscript, see e.g. line 181.

Response: Thank you for pointing this out. We have made the main corrections to the Results, Discussion, and Conclusion sections.

Reviewer 2 Report

The authors have adequately addressed the referee's concerns.

Author Response

Thank you for your wonderful peer review.

Reviewer 3 Report

Thank you for the opportunity to review the second version of the manuscript titled “Physical activity and its diurnal fluctuations vary by non-motor symptoms in patients with Parkinson’s disease: an exploratory study.”

The changes performed by the authors have improved significatively the quality of the manuscript. Thank you for addressing my comments.

However, I have minor comments.

In line 37 punding is a relatively recently discovered feature of Parkinson's disease (PD)  and is a Impulsive-compulsive behavior (ICB).

In (PD) suggests a combination of impulse control disorders (ICDs), such as pathological gambling, hypersexuality, compulsive eating, excessive buying, and compulsive behaviors, such as punding, dopamine dysregulation syndrome (DDS), hoarding, and hobbyism. Hypersexuality and gambling are common in male patients while compulsive buying is common in women patients. The prevalence of punding is not clear and some authors found to vary greatly (between 0.34 to 14%), although there were large disparities in study (1). I suggest to mention the most common side effects of the medication or even the most common ICDs, instead of just one and very uncommon by itself. 

On line 108 NMS is plural, so please change “was”

Thank you for answering my comment about sentence on lines 151-153 and the reference # 45. However, I believe the explanation should be in the body of the document to clarify this concept.

References

  1. Spencer, Ashley H., et al. "The prevalence and clinical characteristics of punding in Parkinson's disease." Movement disorders4 (2011): 578-586.

Author Response

To Reviewer #3

Comments: In line 37 punding is a relatively recently discovered feature of Parkinson's disease (PD) and is a Impulsive-compulsive behavior (ICB).

In (PD) suggests a combination of impulse control disorders (ICDs), such as pathological gambling, hypersexuality, compulsive eating, excessive buying, and compulsive behaviors, such as punding, dopamine dysregulation syndrome (DDS), hoarding, and hobbyism. Hypersexuality and gambling are common in male patients while compulsive buying is common in women patients. The prevalence of punding is not clear and some authors found to vary greatly (between 0.34 to 14%), although there were large disparities in study (1). I suggest to mention the most common side effects of the medication or even the most common ICDs, instead of just one and very uncommon by itself.

Response: Thank you for pointing this out. As you said, it would be inappropriate to limit the description to punding alone, as we have included pulse control and related disorders (ICRDs), excessive daytime sleepiness (EDS), etc., as well as dopaminergic disorders. The text has been revised to say that it may be exacerbated by medications.

“In contrast, some NMS such as psychosis, impulse control and related disorders (ICRDs), excessive daytime sleepiness (EDS), or constipation can also be worsened or even induced by dopaminergic medications [7].” (Line36-)

Comments: On line 108 NMS is plural, so please change “was”

Response: Thank you for pointing this out." We have changed it to "were".

Comments: Thank you for answering my comment about sentence on lines 151-153 and the reference # 45. However, I believe the explanation should be in the body of the document to clarify this concept.

Response: Thank you very much for your accurate point. We have stated as a limitation that the step threshold was calculated from the reference values of the physical activity guidelines by the World Health Organization, but no consensus has been reached.

“Fourth, the 4200 steps/day threshold is an estimate derived from the World Health Organization's physical activity guidelines (adults with PD and other disabilities are recommended to participate in at least 150 minutes/week of moderate-intensity physi-cal activity) [61], but should be interpreted with caution because there is not enough consensus.” (Line 311-)

Ref: Bull, F.C.; Al-Ansari, S.S.; Biddle, S.; Borodulin, K.; Buman, M.P.; Cardon, G.; Carty, C.; Chaput, J.-P.; Chastin, S.; Chou, R.; et al. World Health Organization 2020 Guidelines on Physical Activity and Sedentary Behaviour. Br. J. Sports Med. 2020, 54, 1451–1462, doi:10.1136/bjsports-2020-102955.